# EDITCAST3D: SINGLE-FRAME-GUIDED 3D EDITING WITH VIDEO PROPAGATION AND VIEW SELECTION

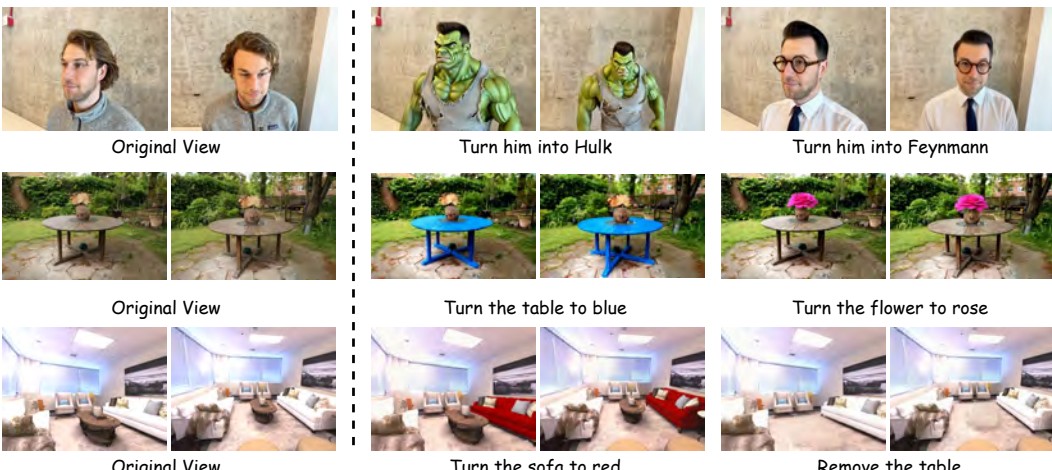

Figure 1: Results of `EditCast3D`. `EditCast3D` enables efficient, detailed, and precise editing of 3D scenes. It requires only a single edited image from the dataset produced by any image editing foundation model as the input. The edit is then propagated across the entire 3D scene. With the addition of the view selection mechanism, `EditCast3D` achieves high-quality reconstructions while demonstrating strong instruction-following and editing capability.

## ABSTRACT

Recent advances in foundation models have driven remarkable progress in image editing, yet their extension to 3D editing remains underexplored. A natural approach is to replace the image editing modules in existing workflows with foundation models. However, their heavy computational demands and the restrictions and costs of closed-source APIs make plugging these models into existing iterative editing strategies impractical. To address this limitation, we propose `EditCast3D`, a pipeline that employs video generation foundation models to propagate edits from a single first frame across the entire dataset prior to reconstruction. While editing propagation enables dataset-level editing via video models, its consistency remains suboptimal for 3D reconstruction, where multi-view alignment is essential. To overcome this, `EditCast3D` introduces a view selection strategy that explicitly identifies consistent and reconstruction-friendly views and adopts feedforward reconstruction without requiring costly refinement. In combination, the pipeline both minimizes reliance on expensive image editing and mitigates prompt ambiguities that arise when applying foundation models independently across images. We evaluate `EditCast3D` on commonly used 3D editing datasets and compare it against state-of-the-art 3D editing baselines, demonstrating superior editing quality and high efficiency. These results establish `EditCast3D` as a scalable and general paradigm for integrating foundation models into 3D editing pipelines.

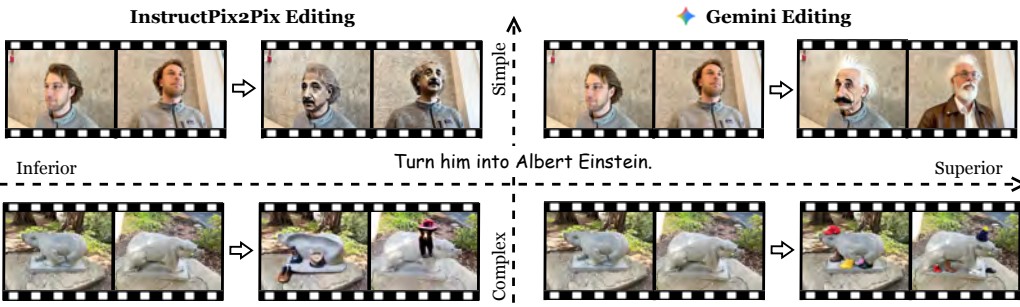

Figure 2: Traditional image editing models like InstructPix2Pix (Brooks et al., 2023) provide relatively consistent edits across views, but their editing capabilities are limited. In contrast, recent foundation models enable more powerful edits, yet often introduce inconsistencies across views.

# 1 INTRODUCTION

3D editing is a fundamental task in many applications such as gaming, film production, and virtual reality. Traditional 3D editing approaches are often based on explicit representations such as meshes or point clouds, but these methods typically require extensive manual effort and tedious parameter tuning. Recent advances have introduced implicit 3D representations, including Neural Radiance Fields (NeRFs) (Haque et al., 2023) and 3D Gaussian Splatting (Chen et al., 2024b; Yan et al., 2024; Bian & Reid, 2025; Chen et al., 2024a; Zhuang et al., 2024; Vachha & Haque, 2024), which offer more flexible and expressive modeling for 3D scenes.

Implicit 3D editing methods originally employ UNet-based diffusion models. Notable examples include Instruct-NeRF2NeRF (Haque et al., 2023), Instruct-GS2GS (Vachha & Haque, 2024), and GaussianEditor (Chen et al., 2024b), all of which build upon the widely adopted 2D diffusion-based image editing model InstructPix2Pix (Brooks et al., 2023). These approaches typically follow an iterative strategy: updating each image in the dataset with the same prompt until the entire set converges to a consistent edit, enabling the reconstruction of an edited 3D scene. However, this iterative design poses high computation cost as the diffusion process is time consuming. More recently, the emergence of the Diffusion Transformer (DiT) architecture (Peebles & Xie, 2023), Flow Matching frameworks (Liu et al., 2022; Lipman et al., 2022; Esser et al., 2024), and unified generation models (Gem; Comanici et al., 2025; Hurst et al., 2024; Deng et al., 2025b) has significantly advanced the field of image editing, leading to the development of powerful foundation models. These innovations have driven remarkable improvements in both image and video generation (Wan et al., 2025; Wu et al., 2025a; Labs et al., 2025). However, despite this progress, the impact on 3D editing has been limited. This gap arises primarily from two challenges: $(i)$ the prohibitive computational or financial cost of incorporating foundation models into existing iterative pipelines, and $(ii)$ the inconsistency of edits across views given the same prompt, as shown in Figure 2. Together, these challenges highlight a disconnect between the rapid progress in 2D editing and the development of 3D editing.

To address these limitations, we propose `EditCast3D`, a 3D editing pipeline that fully leverages recent advances in image and video foundation models. At its core, `EditCast3D` introduces two key components: ❶ First-frame-guided video editing (Section 3.2). During training, masks are applied to the regions requiring edits, and the model learns to reconstruct the masked content by leveraging information from the first frame. This teaches the model to use first-frame guidance effectively. At inference time, only the first frame needs to be edited; the resulting changes are then propagated to the remaining frames via a video generation model. This design drastically reduces the cost of invoking expensive image editing models while mitigating inconsistencies across views. Although propagation yields visually consistent edits, its fidelity can still fall short for high-quality 3D reconstruction. To overcome this we design ❷ a view selection mechanism (Section 3.3). We employ pose-free 3D Gaussian Splatting (Fu et al., 2024) to assess reconstruction quality and identify easy-to-fit views with high PSNR. These selected sparse views are then passed to a feedforward reconstruction pipeline (Fan et al., 2024), resulting in consistent and high-quality 3D reconstructions.

Our contributions are summarized as follows:

- We introduce `EditCast3D`, a pioneering framework that incorporates image editing and video generation foundation models to achieve efficient and high-quality 3D editing.

- We leverage first-frame-guided video editing, which requires only a single edited frame to guide the editing of the entire dataset, thereby achieving high consistency while avoiding the prohibitive cost of iterative editing.

- We further design a view selection mechanism that filters out inconsistent views to enable better reconstruction quality.

- We conduct extensive experiments on commonly used 3D editing datasets, demonstrating the effectiveness of our method in delivering superior visual quality and efficiency.

## 2 RELATED WORK

**3D Editing.** Recent advances in 3D editing have primarily focused on Neural Radiance Fields (NeRFs) (Mildenhall et al., 2020) and 3D Gaussian Splatting (3DGS) (Kerbl et al., 2023). A prevalent approach is to leverage 2D diffusion models as priors: edits are first applied in the 2D image domain and then propagated into the 3D representation. Instruct-NeRF2NeRF (Haque et al., 2023) follows this strategy by using InstructPix2Pix (Brooks et al., 2023) for 2D edits and transferring the results to NeRF. GaussianEditor (Chen et al., 2024b) extends this idea by locating Gaussian splats to be updated via 2D inpainting diffusion, enabling more localized 3D modifications. However, such 2D-to-3D paradigms remain limited. Because edits are iteratively refined in 2D and then consolidated in 3D, they suffer from view inconsistency and scale poorly to dense novel view synthesis due to computational and capacity constraints (Huang et al., 2025). Recent methods such as Edit360 (Huang et al., 2025) have begun to address this challenge by leveraging video diffusion models. Our work, `EditCast3D`, builds on this propagation concept but introduces a more efficient first-frame-guided approach, combined with a novel view-selection mechanism, to explicitly address the reconstruction quality issues faced by propagation-only methods.

**Foundation Models for Image Editing.** The field of image editing has been revolutionized by powerful foundation models capable of understanding and executing complex, instruction-based edits. Unified models like DreamOmni (Xia et al., 2025) and UniWorld-V1 (Zhang et al., 2025) merge generation and editing into a single framework, while others like Step1X-Edit (Liu et al., 2025b) and Draw-In-Mind (Zeng et al., 2025) demonstrate sophisticated intent comprehension. Furthermore, large-scale multimodal foundation models have further pushed the boundaries of image editing and generation. Bagel (Deng et al., 2025a), pretrained on trillions of tokens spanning text, image, video, and web data, supports broad multimodal understanding and generation. Qwen Image (Wu et al., 2025a) achieves notable improvements in complex text rendering and high-fidelity editing. Janus Pro (Chen et al., 2025) advances both multimodal reasoning and text-to-image instruction following, while improving generation stability. Some closed-source models such as Gemini (DeepMind, 2025) and GPT-4o (OpenAI, 2025) demonstrate state-of-the-art capabilities across text, vision, and voice modalities. Despite their power, directly applying these models to each view in a 3D dataset, as illustrated in our Figure 2, often leads to significant inconsistencies. Furthermore, the high computational demand or API costs associated with these models make them impractical for traditional, iterative 3D editing pipelines that require thousands of individual editing steps. `EditCast3D` is designed specifically to harness the advanced editing capabilities of these foundation models while circumventing their high cost and inconsistency issues through a non-iterative, propagation-based pipeline.

**Video Editing.** Modern video editing techniques (Cong et al., 2023; Wu et al., 2023; Kara et al., 2024; Liu et al., 2024) have made significant strides in maintaining temporal consistency, a challenge analogous to achieving multi-view consistency in 3D editing. State-of-the-art methods like VideoDirector (Wang et al., 2025b) and FADE (Zhu et al., 2025) now leverage pretrained text-to-video diffusion models to ensure that edits are propagated coherently across frames. These models are often guided by various signals, such as object features in DIVE (Huang et al., 2024) or user sketches in SketchVideo (Liu et al., 2025a). The development of large-scale, instruction-based datasets like InsViE-1M (Wu et al., 2025b)) has been crucial for training these powerful models. `EditCast3D` adapts this video editing paradigm to the 3D domain by employing a novel form of first-frame

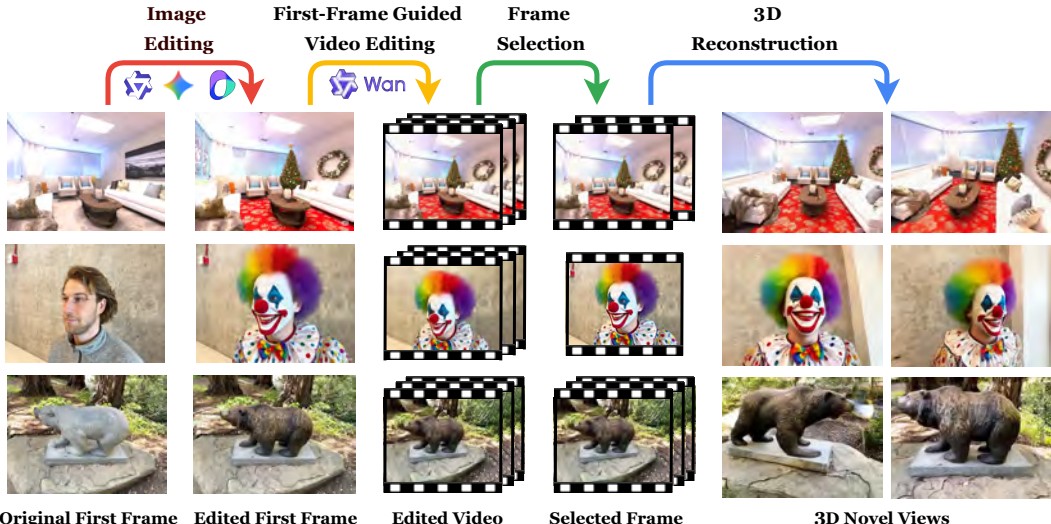

**Original First Frame**    **Edited First Frame**    **Edited Video**    **Selected Frame**    **3D Novel Views**

Figure 3: Starting from a collection of input views, only the first frame is edited using an image editing foundation model. The edit is then propagated to the entire dataset via a video generation foundation model, enabling dataset-level editing with minimal computational cost. Since editing propagation alone may produce inconsistent views, `EditCast3D` further incorporates a view selection strategy that filters for consistent and reconstruction-friendly views. The selected views are finally fed into a feedforward 3D reconstruction module, producing edited 3D assets with high quality and consistency.

guided video generation, where an edit applied to a single view serves as the guidance to consistently transform an entire sequence of views for 3D reconstruction.

## 3 METHOD

### 3.1 OVERVIEW OF EDITCAST3D

In this section, we present an overview of our method. As illustrated in Figure 3, given the original first frame $\mathbf{I}_1$ from an input video $\mathbf{V} = \{\mathbf{I}_i\}_{i=1}^n$ that depicts the original scene $\mathcal{S}$, the user first applies an image editing foundation model to obtain an edited first frame $\tilde{\mathbf{I}}_1$. This step enables complex or prompt-based edits that are difficult to achieve with traditional 3D editing pipelines. We then adopt first-frame-guided video editing, which propagates the edits from $\tilde{\mathbf{I}}_1$ to the entire sequence, producing an edited video $\tilde{\mathbf{V}} = \{\tilde{\mathbf{I}}_i\}_{i=1}^n$ (see Section 3.2). While $\tilde{\mathbf{V}}$ exhibits high visual consistency across frames, not all views are suitable for high-quality 3D reconstruction. To address this, we introduce a view selection mechanism as the third

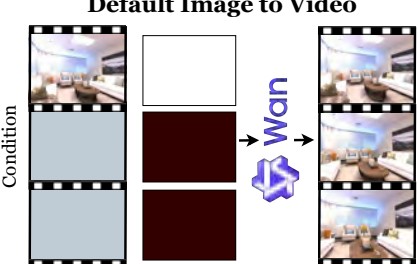

**Default Image to Video**

Figure 4: Default image-to-video generation. Given a masked video sequence and masks, the model fills in the missing regions to synthesize complete frames.

stage that filters out inconsistent or dynamic views, resulting in a subset $\tilde{\mathbf{V}}'$ containing consistent, reconstruction-friendly views, as including inconsistent views would otherwise degrade the geometry and texture quality of the reconstructed scene. Finally, $\tilde{\mathbf{V}}'$ fed into a feedforward 3D reconstruction module at the fourth stage to generate the final edited 3D scene (see Section 3.3).

### 3.2 FIRST-FRAME-GUIDED EDITING OF THE DATASET

**Image-to-Video Generation Model.** Image-to-video generation model employs the first frame as the guidance to generate the subsequent frames. The input are two parts: $(i)$ a masked video to fill with the first frame fully valid and rest of frames masked (grey regions in Figure 4). $(ii)$ masks that

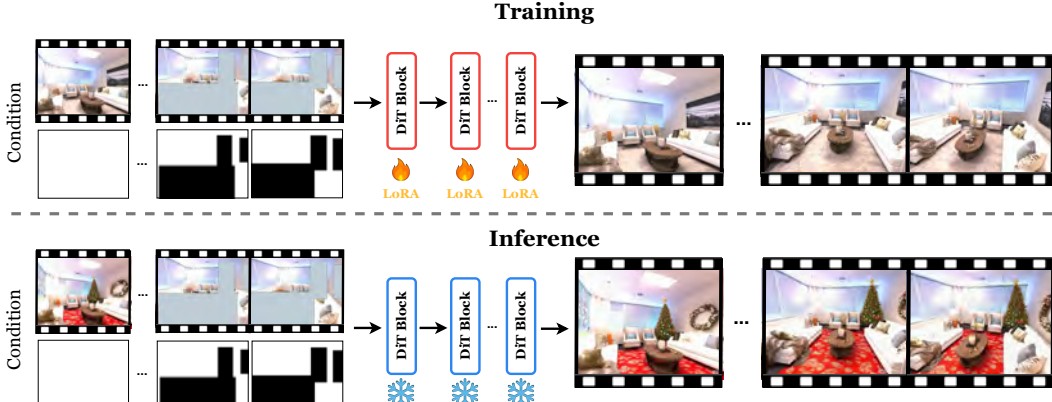

Figure 5: During training, a mask is applied to the regions requiring editing, and LoRAs are trained by reconstructing these masked regions given the first frame as reference. At inference time, the same mask is provided along with the edited first frame. The model then uses the first frame to guide the filling of the masked regions across subsequent frames, thereby achieving consistent video editing.

denote the information validity for the model to use, where only the first frame is fully valid in the default generation configuration (black and white masks in Figure 4). The model then leverages these inputs to fill the frames to generate, producing a video clip.

**Enable First-Frame Editing Guidance.** First-frame-guided video editing requires the model to identify the edited region in the first frame while keeping the untouched regions consistent across frames. To achieve this, we draw inspiration from Gao et al. (2025) to teach the model how to separate the background regions that should be kept versus the regions require editing. Specifically, we leverage the spatiotemporal masks in video generation model (Wan et al., 2025) and employ them to preserve background content while allowing modifications in regions requiring edits. As shown in Figure 5, during the training, the white regions of the masks denote the available information for the model to use. The model is trained to leverage the fully valid first frame and background in subsequent frames to fill the masked regions of in the frames (grey regions in Figure 5). In this way, the model ❶ learns to retain the background and ❷ uses guidance from the first frame to generate consistent edits in following frames. At the inference time, the edited first frame is provided, the fine-tuned model can propagate the edits to the entire sequence. In practice, we apply LoRA fine-tuning (Hu et al., 2022) to adapt the video generation model for this first-frame-guided video editing task (see Figure 5).

Formally, given an input video with $n$ frames $\mathbf{V} = \{\mathbf{I}_1, \ldots, \mathbf{I}_n\}$, we first edit the initial frame using an image editing foundation model to obtain $\tilde{\mathbf{I}}_1$. This frame, together with the original $\mathbf{I}_1$, is compared and used to derive a mask $\mathbf{M}_1$ based on the major different regions. The mask is then propagated to all frames via Segment Anything Model 2 (SAM 2) (Ravi et al., 2024), resulting in $\mathbf{M} = \{\mathbf{M}_1, \ldots, \mathbf{M}_n\}$ and masked video $\mathbf{V}_{\text{mask}}$, with the first-frame mask reset to fully valid ($\mathbf{M}_1 = 1$, all-white in Figure 5). During fine-tuning, LoRA adapters are applied to each layer of the video gener-

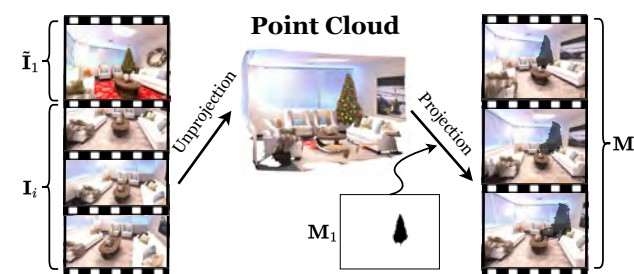

Figure 6: Given an edited first frame with an added object and left untouched frames, we leverage $\pi^3$ (Wang et al., 2025a) to predict the point cloud of the scene. The points corresponding to the added object are then projected onto the remaining views, and the bounding box of the projected pixels is used to generate masks for subsequent editing. **Masks $\mathbf{M}_i$ are overlapped on original images for visualization.**

ation model to enable the first-frame-guided editing task. The model is trained with the standard diffusion loss to reconstruct the noise:

$$\mathbf{x}_t = \text{AddNoise}\left(\mathcal{E}(\mathbf{V}), \epsilon, t\right), \qquad \mathcal{L} = \mathbb{E}_{t,\epsilon}[\|\epsilon_\theta(\mathbf{x}_t, t; \mathbf{V}_{\text{mask}}, \mathbf{M}) - \epsilon\|_2^2]. \qquad (1)$$

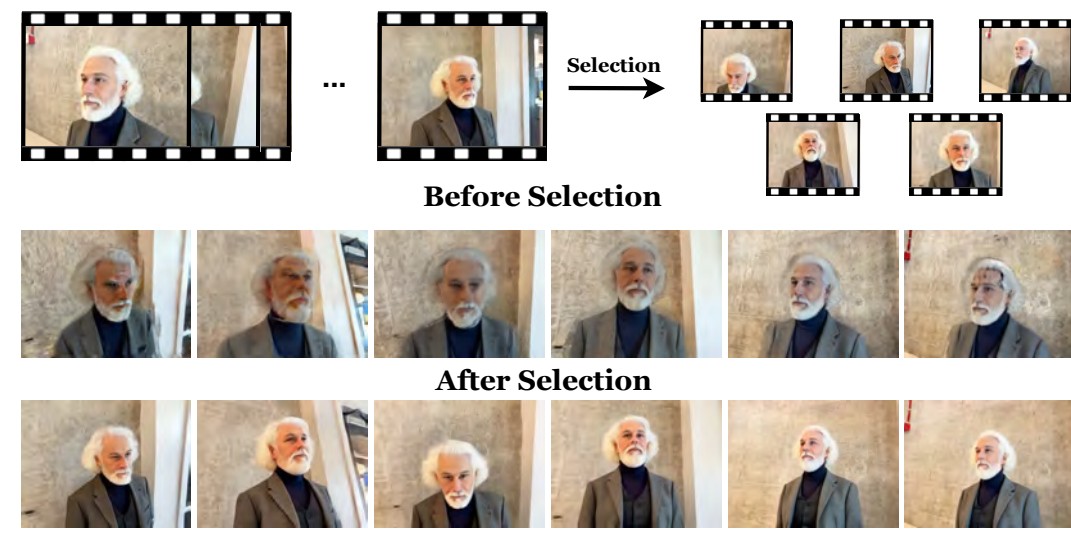

Figure 7: To mitigate the inconsistency issue, `EditCast3D` first employs CF-3DGS (Fu et al., 2024), a pose-free 3D Gaussian Splatting reconstruction method, to obtain an initial reconstruction of the scene. Training views with higher PSNR are then identified as more consistent views and selected. These selected views are subsequently passed to InstantSplat (Fan et al., 2024) to perform the final feedforward reconstruction for improved consistency and quality.

**Mask for Adding Objects.** A challenge of the above framework is generating masks for newly added objects, since SAM 2 can only track objects already present in the video. To address this, we leverage the off-the-shelf point cloud prediction model $\pi^3$ (Wang et al., 2025a) (Figure 6). Specifically, we concatenate the edited first frame with the subsequent unedited frames as $\mathbf{V}_{\mathrm{mix}} = \{\tilde{\mathbf{I}}_1, \mathbf{I}_2, \ldots, \mathbf{I}_n\}$, and feed them into $\pi^3$ to predict a dense pixel-aligned point cloud $\mathbf{P}_{\mathrm{mix}}$ for the entire scene. Owing to the strong capability of $\pi^3$, the model aggregates information across frames to recover the scene geometry while also placing the added object from $\tilde{\mathbf{I}}_1$ at the correct 3D position. Since the mask of the added object $\mathbf{M}_{\mathrm{add}}$ is available in the first frame, we select the corresponding pixels and their 3D points from $\mathbf{P}_{\mathrm{mix}}$. These 3D points are then projected onto the remaining frames, where their bounding boxes are used to define the masks of the added object in all views.

### 3.3 VIEW SELECTION TO ENHANCE RECONSTRUCTION QUALITY

**View Selection.** Although the above first-frame-guided video editing enables high-quality dataset editing, its 3D consistency can be suboptimal, leading to artifacts and blur in rendered images (Figure 7). To prevent reconstruction degradation, we introduce a view selection mechanism that leverages pose-free 3D Gaussian Splatting to assess per-view consistency.

Given the edited sequence $\tilde{\mathbf{V}} = \{\tilde{\mathbf{I}}_i\}_{i=1}^n$, we run CF-3DGS (Fu et al., 2024) on the entire set in a pose-free manner, co-optimizing camera poses and radiance. After convergence, we render the training views as $\hat{\mathbf{V}} = \{\hat{\mathbf{I}}_i\}_{i=1}^n$ and compute a per-view difficulty score

$$\mathrm{score}_i = \lambda_1 \big\|\hat{\mathbf{I}}_i - \tilde{\mathbf{I}}_i\big\|_2^2 + \lambda_2\big(1 - \mathrm{SSIM}(\hat{\mathbf{I}}_i, \tilde{\mathbf{I}}_i)\big) + \lambda_3\,\mathrm{LPIPS}(\hat{\mathbf{I}}_i, \tilde{\mathbf{I}}_i), \qquad (2)$$

where lower values indicate easier-to-fit (i.e., more consistent) views. We retain views whose scores are below a threshold:

$$\tilde{\mathbf{V}}' = \Big\{\tilde{\mathbf{I}}_i \mid \mathrm{score}_i \leq \tau \Big\}. \qquad (3)$$

If from the above step $|\tilde{\mathbf{V}}'| < K$, we instead keep the $K$ lowest-score views:

$$\tilde{\mathbf{V}}' = \Big\{\tilde{\mathbf{I}}_i \mid i \in \mathrm{argsort}_K\big(\{\mathrm{score}_i\}_{i=1}^n\big)\Big\}. \qquad (4)$$

**Feedforward Reconstruction.** Finally, we feed the selected views and their poses into a feedforward 3D reconstruction model, InstantSplat (Fan et al., 2024), to obtain the final edited scene:

$$\tilde{\mathcal{S}} = \mathrm{InstantSplat}(\tilde{\mathbf{V}}'). \qquad (5)$$

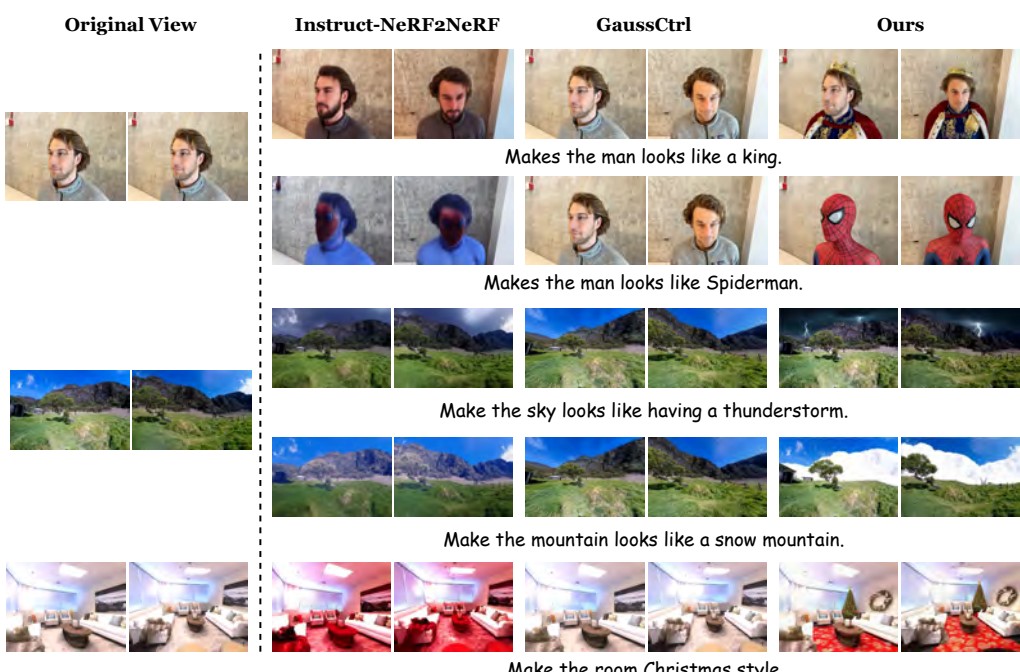

Figure 8: Comparison of `EditCast3D` with Instruct-NeRF2NeRF and GaussCtrl. Across object-centric, outdoor, and room-scale scenes, **`EditCast3D` consistently achieves higher-quality editing with better prompt adherence, while baseline methods often fail under challenging prompts.**

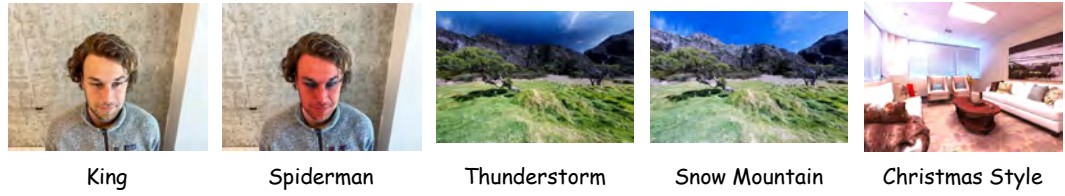

Figure 9: Examination of edited image from baseline methods. Given the low-quality 3D editing of baselines, we further check the edited images from their pipeline. **Results show that the underlying image editing mdoel is insufficient to understand and follow the complex prompts.**

This two-stage design of view selection followed by feedforward reconstruction improves geometric fidelity and visual consistency while avoiding costly iterative refinements.

## 4 EXPERIMENT

Building on the framework described above, we conduct a comprehensive evaluation of `EditCast3D` against SOTA iterative 3D editing baselines. We compare to Instruct-NeRF2NeRF (Haque et al., 2023) (NeRF-based) and GaussCtrl (Wu et al., 2024) (3DGS-based). Our benchmarks cover object-centric scenes using the IN2N benchmark introduced by Instruct-NeRF2NeRF, large-scale/outdoor scenes from Mip-NeRF 360 (Barron et al., 2022), and indoor room-scale scenes from Replica (Straub et al., 2019). Unless otherwise noted, `EditCast3D` uses `Gemini-2.5-Flash` (Comanici et al., 2025) as the image-editing backbone. We first assess editing quality and efficiency in Section 4.1, then analyze multi-view/3D consistency under video-based propagation in Section 4.2, and finally showcase downstream applications enabled by our pipeline in Section 4.3.

Table 1: Efficiency comparison of `EditCast3D` with baseline methods. We report editing time and memory consumption for a single scene under the default hyperparameter settings. **`EditCast3D` offers shorter editing time at the cost of similar memory consumption.**

| | Face | | Farm | | Room | |
|---|---|---|---|---|---|---|
| | Time (h) | Memory (GB) | Time (h) | Memory (GB) | Time (h) | Memory (GB) |
| Instruct-NeRF2NeRF | 3.6 | 10.9 | 4.4 | 15.1 | 5.0 | 19.5 |
| GaussCtrl | 3.3 | **6.8** | 3.9 | **11.2** | 4.8 | **15.5** |
| Ours | **1.6** | 16.7 | **2.3** | 20.9 | **3.7** | 21.4 |

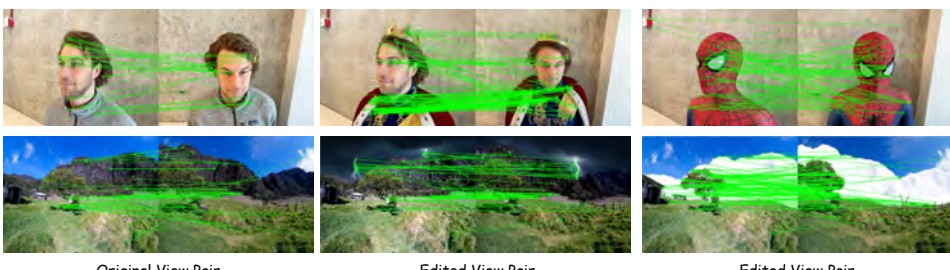

| Original View Pair | Edited View Pair | Edited View Pair |
|---|---|---|

Figure 10: Matching results comparison. Green lines represent matched pairs under the threshold. More matches indicate a higher level of geometric consistency between the two views. **Our method achieves similar amount of matched pairs to the original views, suggesting high consisitency.**

## 4.1 EDITCAST3D ACHIEVES SUPERIOR 3D EDITING QUALITY AND EFFICIENCY

As shown in Figure 8, across object-centric, outdoor, and indoor settings, `EditCast3D` consistently delivers superior qualitative editing results. ❶ On the object-centric and outdoor datasets, by leveraging strong image editing foundation models, `EditCast3D` produces prompt-faithful, fine-grained, and spatially localized edits with higher fidelity and tighter control than the baselines. `EditCast3D` not only reliably interprets appearance-specific attributes (e.g., "king", "Spider-Man") and applies semantically appropriate modifications, but also accurately and robustly localizes the "sky" and "mountain" regions, restricting the edit to these areas. In contrast, baseline methods often yield incomplete or malformed subject insertions (e.g., partial "Spider-Man" or "king" attributes), as observed for Instruct-NeRF2NeRF. ❷ `EditCast3D` also handles vague prompts effectively (see the last row of Figure 8): it modifies only the Christmas tree, wreaths (flower rings), and carpet while preserving the scene layout and global chromatic tone. Baselines, however, tend to misinterpret the prompt—shifting global colors and failing to localize edits at the scene level. ❸ We trace these failures primarily to structural limitations of the underlying 2D image-editing backbones driving iterative pipelines. As shown in Figure 9, the editors used by the baselines systematically struggle to reliably execute semantically complex prompts, producing low-fidelity 2D edits; when applied to 3D editing, the resulting 3D reconstructions inherit analogous artifacts and degradation (Figure 8).

Beyond visual quality, `EditCast3D` is substantially more efficient than iterative baselines (Table 1). We report average end-to-end editing time across two canonical edit prompts—"Face" and "Farm"—measured under identical hardware, resolution, and batch-size configurations. Pipelines such as Instruct-NeRF2NeRF and GaussCtrl update each view over multiple refinement rounds and repeatedly invoke the diffusion-based InstructPix2Pix to edit every image in the dataset; the cumulative diffusion steps dominate wall-clock time and overall pipeline latency. In contrast, `EditCast3D` performs a single image-editing call on the first frame, followed by a LoRA-based fine-tuning of the video generation model and a lightweight view-selection stage, reducing editor invocations from $O(N \times R)$ to $O(1)$ (where $N$ is the number of views and $R$ the number of refinement rounds) and avoiding repeated diffusion entirely. It is worth noting that the memory consumption difference of the baseline methods is primarily by the 3D representations used (NeRF vs. 3DGS), rather than the image editing model, while `EditCast3D` introduces slightly more memory usage due to the video generation model fine-tuning process. As a consequence, `EditCast3D` delivers up to 50% reduction in end-to-end runtime while maintaining a comparable peak GPU memory footprint.

Table 2: Number of matched keypoint pairs for original and edited view pairs. "King"-"Spiderman" and "Thunderstorm"-"Snow Mountain" share the same original view pairs. **EditCast3D achieves high 3D consistency, with a comparable number of matched pairs to the originals.**

|  | King | Spiderman | Thunderstorm | Snow Mountain | Christmas Style |
|---|---|---|---|---|---|
| Original |  | 139 |  | 68 | 283 |
| Edited | 163 | 96 | 79 | 155 | 259 |

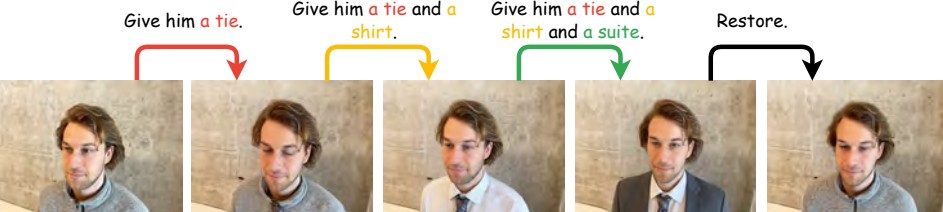

Figure 11: Progressive editing of the same dataset and restoration to the original view.

## 4.2 EDITCAST3D HAS HIGH 3D CONSISTENCY

We further evaluate the geometric consistency of the scenes edited by EditCast3D, since our framework relies on video editing and lacks an intrinsic 3D consistency guarantee. Specifically, we adopt a feature matching algorithm (Rublee et al., 2011) to establish correspondences across a pair of views selected from Figure 8. The edited pairs are then compared against the original unedited views, which are inherently 3D-consistent and thus serve as an upper bound for evaluation.

The visualization of matching results is presented in Figure 10, and the number of matched pairs is reported in Table 2. From the visualizations, EditCast3D exhibits a comparable number, density, and spatial distribution of matched pairs to those of the original view pairs. Beyond qualitative similarity, Table 2 shows that the edited pairs produced by EditCast3D achieve a similar number of reliable matches compared to the original views, indicating that our first-frame-guided video editing design, combined with the view selection mechanism, effectively preserves geometric consistency.

## 4.3 EDITCAST3D ENABLES VARIOUS APPLICATIONS.

In this section, we present two examples of EditCast3D's editing to highlight its strengths. We consider two scenarios: $(i)$ progressive editing, where a second edit is applied on top of the first, requiring the model to balance multiple edits while preserving existing visual features; and $(ii)$ editing restoration, which evaluates the model's ability to maintain the original scene identity. As shown in Figure 11, EditCast3D achieves high-quality progressive editing for incremental additions and can also restore the original view when given the first unedited frame.

## 5 CONCLUSION

In this work, we introduced EditCast3D, a novel 3D editing pipeline that integrates image editing and video generation foundation models through first-frame-guided propagation and view selection. By requiring only a single edited frame, our approach significantly reduces reliance on costly iterative editing while mitigating inconsistencies that arise from independent per-view edits. The subsequent view selection stage further improves geometric fidelity by filtering out inconsistent views and enabling efficient feedforward reconstruction. Extensive experiments demonstrate that EditCast3D achieves superior editing fidelity, stronger instruction alignment, and improved efficiency compared to SOTA baselines. We believe EditCast3D represents a step toward bridging the gap between recent advances in foundation models and practical 3D editing.

## ETHICS STATEMENT

We adhere to the ICLR Code of Ethics. Our work focuses on developing a 3D editing framework that builds upon existing open-source image, video, and 3D datasets, as well as publicly available foundation models. No private, sensitive, or personally identifiable data are involved. The goal of this research is to advance controllable and efficient 3D editing, and it does not raise foreseeable ethical concerns or produce harmful societal outcomes.

## REPRODUCIBILITY STATEMENT

Reproducibility is central to our work. All datasets used in our experiments are standard benchmarks that are publicly available. We provide full details of the training setup, model architectures, and evaluation metrics in the main paper and appendix. Upon acceptance, we will release our codebase, including scripts for preprocessing, training, and evaluation, along with configuration files and documentation to facilitate exact reproduction of our results. Random seeds and hyperparameters will also be included to further ensure reproducibility.

## THE USE OF LARGE LANGUAGE MODELS (LLMS)

To enhance clarity and readability, we employed OpenAI's GPT-5 and GPT-5-thinking models exclusively as language polishing tools. Their role was limited to proofreading, grammatical correction, and stylistic refinement—functions comparable to those of conventional grammar checkers and dictionaries. These tools did not contribute any new scientific content or ideas, and their usage is consistent with standard practices in manuscript preparation.

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
