# OpenReview forum: "$\texttt{EditCast3D}$: Single-Frame-Guided 3D Editing with Video Propagation and View Selection"
_ICLR.cc/2026/Conference — ICLR 2026 Conference Withdrawn Submission_

### Official Review · Reviewer_4aG8 · 2025-10-31

**Soundness:** 1
**Presentation:** 1
**Contribution:** 1
**Rating:** 0
**Confidence:** 5

**Summary:**

This paper studies text-guided 3D scene editing. The idea of the method is to render the scene into a video, edit the first frame with powerful VLMs, propagate the frame editing throughout the whole video to obtain the edited video, and select frames to reconstruct back to the scene. The proposed method is claimed to outperform the only two baselines.

**Strengths:**

- The method is easy to understand.
- Qualitative results are provided in this paper across different editing prompts.

**Weaknesses:**

- **Crucial.** The result presentation is highly problematic.
    - All the images in this paper are at low resolution, showing clear grids when zooming in. It is unable to see any details.
    - No video is provided to holistically view the whole 3D scene to verify its quality.
    - Also, there are no quantitative evaluations at all in this paper. Tab.1 is about efficiency, and Tab.2 is about keypoint matching, but none of the baselines are mentioned.
    - Therefore, the results do not convince me at all.
- **Crucial.** The baseline selection and comparison are highly problematic.
    - The only baselines are Instruct-NeRF2NeRF and GaussCtrl, which are neither the latest nor SOTA. There are many 3D editing papers, including EN2N, CSD, PDS, ViCA-NeRF, GaussianEditor, ProEdit, DreamEditor, etc. Some of them have even been discussed in related work, but none have been compared.
    - The proposed method also includes a new first-frame-guided video editing method, which is also a well-studied topic in video editing. Existing work includes VideoShop, Slicedit, BIVDiff, StableV2V, AnyV2V, etc.. Most of them are even training-free, unlike the proposed method. It is still necessary to compare all the baselines that replace the proposed video editing with one of these.
- The proposed video editing method is neither efficient nor general. It not only requires training a large video diffusion model but also requires a mask of the edited part (which disables global style editing). Most of the methods mentioned above are mask-free and even training-free. The proposed methods show no superiority compared with those baselines.
- Despite the low-resolution presentation, there are still many issues in the results.
    - In Fig.1 (IN2N-face), "Turn him into Hulk" mistakenly changes the size and position of the head and body, while "Turn him into Feynmann" mistakenly changes the clothes.
    - In Fig.8, "Make the room Christmas style" should be a global style transfer. However, only the floor is re-colored, and a Christmas tree is added.
- The "pose-free reconstruction" method for selection might be unstable. When the video contains (unexpected but usual) motion, the reconstruction may be random and unreliable. Depending on such a model may result in unstable performance in different random seeds.

**Questions:**

- Could you please provide the video results and the missing baseline results?
- I noticed an arXiv paper which also uses video diffusion for scene editing: V2Edit. Could you please evaluate whether the proposed method can perform the editing tasks on V2Edit's website, e.g., add a parrot on the shoulder?

---

### Official Review · Reviewer_Kahs · 2025-10-31

**Soundness:** 2
**Presentation:** 2
**Contribution:** 4
**Rating:** 4
**Confidence:** 5

**Summary:**

This paper introduces EditCast3D, a novel 3D editing pipeline that leverages recent advances in foundation models for image and video generation. The framework explores the potential of video generation model in 3D editing by converting the 3D editing task into a video editing task. Specifically, it enables first-frame video editing pipeline and a view selection mechanism to select frames that are consistently edited.

**Strengths:**

1. The manuscript is well organized and easy to follow. The formulas and methodological details are clearly presented, making the technical contributions easy to understand.
2. The first-frame editing pipeline is a well-established paradigm in video editing, and I appreciate the exploration of extending this idea to 3D editing.
3. The presented editing results are promising and appear to be on par with the state-of-the-art performance of current 3D editing methods.

**Weaknesses:**

My main concerns lie in the experimental part:

1.Regarding the view-selection mechanism, the core idea of the paper is that if the edited frames remain consistent with the source frame according to several similarity metrics, then the editing is considered not to have damaged the 3D consistency and structure. However, I am not convinced that SSIM, LPIPS, and per-pixel distance alone can achieve a reliable and generalizable filtering. For editing cases with larger appearance or geometric changes, especially those involving slight geometry modifications, I believe that consistent and inconsistent edited frames cannot be effectively distinguished using such simple image similarity metrics from many viewpoints. In the qualitative ablation study, the authors only show a single example of a human face, which is far from sufficient to convince me that the view-selection mechanism is truly effective rather than a cherry-picked case. I strongly recommend that the authors present more results in the revised version to demonstrate that the observed improvements are indeed due to the proposed view-selection mechanism, rather than the intrinsic 3D awareness capability of the video generation backbone.

2.The presented figures include too few viewpoints to fully assess multi-view consistency. For top-tier conference 3D editing research, rendered video results are crucial to verify 3D coherence and rule out cherry-picking. The absence of supplementary videos showing the raw video edits, the filtered results after view selection, and the final 3D reconstructions makes it difficult to evaluate the true consistency and effectiveness of the approach.

**Questions:**

See weaknesses. I would suggest the authors respond to the listed concerns. Besides, I am also curious why the authors choose to use a LoRA module to preserve the background. After obtaining the dynamic editing region masks with SAM, wouldn’t it be sufficient to directly blend the unmasked regions with the corresponding areas from the source frames? Moreover, the codebase of GaussianEditor provides a gradient mask mechanism for background preservation — wouldn’t that be a more efficient and straightforward solution?

---

### Official Review · Reviewer_mANf · 2025-10-31

**Soundness:** 3
**Presentation:** 2
**Contribution:** 2
**Rating:** 4
**Confidence:** 3

**Summary:**

This paper proposes EditCast3D, a new framework for 3D object editing. To overcome the high computational cost and inconsistency issues when applying image-based foundation models to 3D editing, EditCast3D employs video generation models to propagate edits across views and introduces a view selection strategy that ensures multi-view consistency and efficient reconstruction. Experiments demonstrate that EditCast3D achieves superior editing quality and efficiency compared to existing 3D editing methods, establishing it as a scalable and general approach for integrating foundation models into 3D editing pipelines.

**Strengths:**

This is a reasonable and well-designed approach. In particular, the first-frame guidance strategy is a logical choice for maintaining view consistency. The qualitative results are also excellent.

**Weaknesses:**

[1] From a novelty perspective, isn’t the proposed approach somewhat lacking? The techniques used here seem largely similar to those already explored in the video editing domain.

[2] Were there any specific challenges encountered when extending 2D editing methods to the 3D domain?

[3] Are there no video-based results provided?

**Questions:**

See above

---

### Official Review · Reviewer_TWqw · 2025-11-01

**Soundness:** 2
**Presentation:** 1
**Contribution:** 2
**Rating:** 2
**Confidence:** 5

**Summary:**

The paper introduces EditCast 3D, a pipeline for 3D scene editing that leverages recent advances in image editing and video generation foundation models. The core problem it addresses is the computational cost and view inconsistency associated with integrating powerful, but expensive, image editing foundation models into traditional iterative 3D editing pipelines (like those based on InstructPix2Pix).

**Strengths:**

1. The paper integrates expensive, state-of-the-art image and video foundation models into a 3D editing pipeline.

**Weaknesses:**

1. Lack of Quantitative Quality Metrics: The claims of "superior editing quality" rely heavily on qualitative comparisons. The paper is missing standard quantitative metrics (e.g., FID, CLIP score) for editing fidelity and prompt adherence, making the claimed superiority impossible to verify objectively.

2. Unconvincing Baselines: The comparison is limited to only two baselines (Instruct-NeRF2NeRF and GaussCtrl). These methods, which use the older InstructPix2Pix backbone, are no longer considered State-of-the-Art, inflating the comparative advantage of the proposed method.

3. Missing Ablation of Key Component: The paper does not include an ablation study for the crucial view selection mechanism. Without a comparison against the pipeline without view selection, the purported contribution of this consistency-restoring step to the final reconstruction quality remains unverified and speculative.

4. Low-Resolution Figures: The visual results presented in the figures (e.g., Figure 1 and 8) are of low resolution and poor quality, making it extremely difficult to verify the fine-grained geometric and texture fidelity claimed by the authors.

5. Unclear Necessity: The motivation claims to overcome the cost of foundation models in iterative pipelines, yet the proposed method is still fundamentally dependent on expensive foundation models (e.g., Gemini-2.5-Flash and Wan) for the core editing and propagation steps.

6. Overstated Pioneering Claim: The claim of pioneering the use of video generation for 3D editing is overstated, as existing contemporary work (e.g., V2Edit [1]) has already explored this direction. The novelty should be narrowed to the specific single-frame-guided non-iterative approach and the selection mechanism.

[1] Zhang Y, Chen J K, Lyu J, et al. V2Edit: Versatile Video Diffusion Editor for Videos and 3D Scenes[J]. arXiv preprint arXiv:2503.10634, 2025.

**Questions:**

Please see the weakness.

---

### Note · Authors · 2025-11-18

I have read and agree with the venue's withdrawal policy on behalf of myself and my co-authors.